# FedLPA: Local Prior Alignment for Heterogeneous Federated Generalized Category Discovery

**Geeho Kim**[1]          **Jinu Lee**[2]          **Bohyung Han**[1,3]

[1]ECE & [3]IPAI, Seoul National University
[2]Toss (Viva Republica)
{snow1234, bhhan}@snu.ac.kr
jinu.lee@toss.im

## Abstract

Federated Generalized Category Discovery (Fed-GCD) aims to train a global model that classifies seen classes while discovering novel ones from data distributed across heterogeneous clients. Existing GCD methods often rely on unrealistic assumptions, such as prior knowledge of the number of novel classes or balanced class distributions across clients. We propose Federated Local Prior Alignment (FedLPA), which eliminates these assumptions by grounding learning in client-specific structures and aligning predictions with locally derived priors. Specifically, each client constructs a similarity graph refined with high-confidence signals from seen classes, and then identifies local concepts and prototypes via Infomap clustering. Building on these discovered structures, we introduce Local Prior Alignment (LPA), a self-distillation mechanism that aligns batch-level predictions with empirical class prior derived from concept assignments. Through iterative local structure discovery and adaptive prior refinement, FedLPA achieves robust generalized category discovery under severe data heterogeneity. Extensive experiments demonstrate that FedLPA significantly outperforms existing federated GCD methods across both fine-grained and standard benchmarks.

## 1   Introduction

Machine learning models increasingly operate in open-world settings where not all classes are available during training. Generalized Category Discovery (GCD) [22, 3] addresses this challenge by categorizing unlabeled data containing instances from both seen and novel classes, leveraging knowledge from the labeled examples. However, existing studies on GCD have predominantly focused on centralized settings, assuming universal access to training data, which comprises two types of classes—seen and novel—with annotations available for only a subset of the seen classes. In such settings, the number of seen classes is known, and the number of novel classes is typically assumed to be given. This centralized formulation, however, overlooks a more practical scenario where both data and computational resources are distributed across multiple clients.

We address the Federated Generalized Category Discovery (Fed-GCD) problem, which extends GCD to the federated learning setting. In this scenario, each local client independently maintains its own training data without sharing it with others due to privacy constraints [15]. While the core objective of Fed-GCD aligns with that of centralized GCD, training within a privacy-preserving federated learning framework introduces additional challenges due to the following reasons. First, each client encounters more severe data heterogeneity and class imbalance [9, 8, 1], as training examples are partitioned across individual clients. Second, and more critically, the sets of classes may differ across clients. In other words, unlike the prevalent assumption in centralized GCD [3, 27, 23, 25, 18, 29], each client observes only a *partial* class set of the global label space; consequently, the number of

novel classes varies across clients and is unknown at the client level, while the total number of novel classes over all clients is also unknown.

Existing centralized GCD approaches often rely on assumptions that are unrealistic in the federated learning setting. First, most methods require prior knowledge of the total number of novel classes [3, 27, 23, 25, 18, 29] to configure their classifiers and loss functions. Second, these methods often assume uniform class distributions. For instance, classifier-based approaches such as SimGCD [27] and its variants [25, 23] employ entropy regularization to encourage balanced predictions across all classes (both seen and novel), while clustering-based methods [20, 16, 30] impose cluster-size balance constraints to ensure well-separated novel class clusters. Even recent federated methods inherit these assumptions; for example, FedoSSL [28] presumes knowledge of the total number of novel classes across all clients, while AGCL [19] similarly assumes uniform cluster distributions. These assumptions directly conflict with the inherent data heterogeneity and class imbalances prevalent in Fed-GCD.

To overcome these limitations, we propose a novel federated learning framework that eliminates such unrealistic assumptions by discovering data structure at the client level. Rather than requiring prior knowledge of the global number of novel classes, our approach enables each client to construct a local similarity graph over its entire local dataset, comprising both labeled and unlabeled data, by incorporating ground-truth labels and high-confidence pseudo-labels assigned to unlabeled examples. Applying Infomap clustering to this graph reveals client-specific class priors including the cardinality of local novel classes.

Building on these discovered concept structures, we introduce Local Prior Alignment (LPA), a novel self-distillation strategy that enhances generalized category discovery under skewed local data by aligning the model's batch-level predictions for unlabeled examples with the locally-derived concept distributions. This simple mechanism effectively guides the model toward each client's true data structure, enabling robust representation learning across heterogeneous clients. The proposed approach demonstrates remarkable performance improvements across all datasets and settings consistently, outperforming existing Fed-GCD baselines by significant margins.

Our main contributions are summarized as follows.

- We propose FedLPA, a novel federated learning framework for generalized category discovery that eliminates the need for prior knowledge of novel class counts while naturally accommodating non-*i.i.d.* data distributions across clients.
- We establish a graph-based category discovery mechanism that constructs local similarity graphs enriched with high-confidence seen-class signals, and applies Infomap clustering to derive client-specific class priors and concept prototypes.
- We introduce Local Prior Alignment (LPA), a self-distillation strategy that adaptively regularizes batch-level predictions by aligning them with empirical local priors, thereby enhancing robustness under severe data heterogeneity.
- FedLPA demonstrates its outstanding performance in terms of robustness to client heterogeneity on fine-grained and standard object recognition benchmarks under various settings.

In the rest of this paper, we first review related works in Section 2 and discuss our main algorithm in Section 3. Section 4 presents our experimental results and Section 5 concludes this paper.

## 2  Related Works

### 2.1  Centralized generalized category discovery

The objective of Generalized Category Discovery (GCD), formulated by [22, 3], aims to classify samples from seen categories while simultaneously discovering novel classes, leveraging knowledge from labeled data. Unlike Novel Class Discovery (NCD) [7], which assumes unlabeled data contains only novel classes, GCD addresses a more realistic and challenging scenario, where unlabeled data includes both known and novel classes. Existing GCD approaches typically adopt two main paradigms: parametric classifier learning and non-parametric representation learning.

Parametric methods [3, 27, 23, 25, 26, 14] build a learnable classifier and optimize it using labeled data and pseudo-labeled data generated from model predictions. These methods incorporate adaptive

margins [3] or entropy regularization [27, 23, 25, 14] to achieve balanced pseudo-labeling, and leverage mean-teacher frameworks [23] or prompt-tuning [25] to refine pseudo-label quality. In contrast, non-parametric methods employ contrastive learning with diverse strategies—combined losses [22], multiple projection heads [6], hierarchical [20] or concept-level [18] formulations, or GMM-based generative sampling [30]—to enhance feature generalization to novel categories. However, these approaches focus on centralized settings, and rely on assumptions ill-suited for realistic federated learning settings. First, they often assume prior knowledge of the number of novel classes [3, 27, 23, 25, 20, 18], or require labeled validation data to estimate the class counts [22, 18, 20, 6]. Second, they frequently make the stronger assumption that class distribution is balanced. For instance, parametric methods [27, 16, 23, 25] commonly adopt mean entropy maximization (ME-MAX [2]) to enforce uniform predictions across all classes, while non-parametric approaches [20, 16, 30] impose cluster-size balance constraints to maintain clusters corresponding to novel classes. Such assumptions are often impractical in real-world distributed settings, where data is partitioned heterogeneously across clients.

## 2.2 Federated generalized category discovery

To address the limitations of centralized approaches, recent work has explored Federated Generalized Category Discovery (Fed-GCD). Fed-GCD extends GCD to the federated learning paradigm [15], where clients collaboratively train a global model without sharing their raw data, thereby preserving privacy. The objective of Fed-GCD is training a global model that discovers novel categories and accurately classifies known categories distributed across heterogeneous clients. This task is more challenging than centralized GCD, as each client observes only a *partial* class set of the global label space due to severe data heterogeneity. Early work, FedoSSL [28] tackles heterogeneity in novel class distributions by distinguishing *locally unseen* classes (novel in some clients) from *globally unseen* classes (novel across all clients). However, FedoSSL assumes the total number of novel classes is known a priori and that seen class data is *i.i.d.* and balanced across clients. More recently, AGCL [19] addresses Fed-GCD under more realistic settings, where both seen and novel classes exhibit highly skewed and non-*i.i.d.* distributions, employing GMM-based contrastive learning to learn robust representations of both seen and novel classes. However, AGCL assumes a uniform cluster prior and samples accordingly, which leads to a mismatch with the true local data distributions. Additionally, it requires clients to send local class representations to the server, raising privacy concerns and incurring additional communication overhead. In contrast, our framework overcomes these limitations through client-level structure discovery and adaptive prior alignment, without assumption of balanced class distributions while preserving privacy.

## 3 Proposed Algorithm: FedLPA

This section presents our approach for federated generalized category discovery, referred to as FedLPA, which combines graph-based local category discovery and adaptive prior regularization.

### 3.1 Problem setup

We consider a federated learning setting with $N$ clients $\mathcal{C} = \{C_n\}_{n=1}^N$. Each client $C_n$ maintains a local dataset $\mathcal{D}_n = \mathcal{D}_n^l \cup \mathcal{D}_n^u$, partitioned into labeled and unlabeled subsets. The labeled subset $\mathcal{D}_n^l = \{(x_i, y_i)\}_{i=1}^{|\mathcal{D}_n^l|}$ contains instances with known labels $y_i \in \mathcal{Y}_n^l$, while the unlabeled subset $\mathcal{D}_n^u = \{x_i\}_{i=1}^{|\mathcal{D}_n^u|}$ contains instances whose true labels belong to $\mathcal{Y}_n^u$ but remain unobserved. The global known label set is defined as $\mathcal{Y}^l = \bigcup_{n=1}^N \mathcal{Y}_n^l$, and the global label space (encompassing both known and novel classes) is $\mathcal{Y}^u = \bigcup_{n=1}^N \mathcal{Y}_n^u$ with $\mathcal{Y}^l \subseteq \mathcal{Y}^u$. Novel classes correspond to $\mathcal{Y}^u \setminus \mathcal{Y}^l$ and appear exclusively in the aggregated unlabeled data $\mathcal{D}^u = \bigcup_n \mathcal{D}_n^u$. The objective of Fed-GCD is to collaboratively train a global model $f : \mathcal{X} \to \mathcal{Y}^u$ from $\{\mathcal{D}_n\}_{n=1}^N$ that accurately classifies all unlabeled instances into their true classes in $\mathcal{Y}^u$. Crucially, client data distributions are heterogeneous—local label sets $\mathcal{Y}_n^l$ and $\mathcal{Y}_n^u$ vary across clients, and the cardinalities $|\mathcal{Y}^u|$ and $|\mathcal{Y}_n^u|$ are unknown a priori. Moreover, raw data remains local to each client and cannot be shared due to privacy constraints.

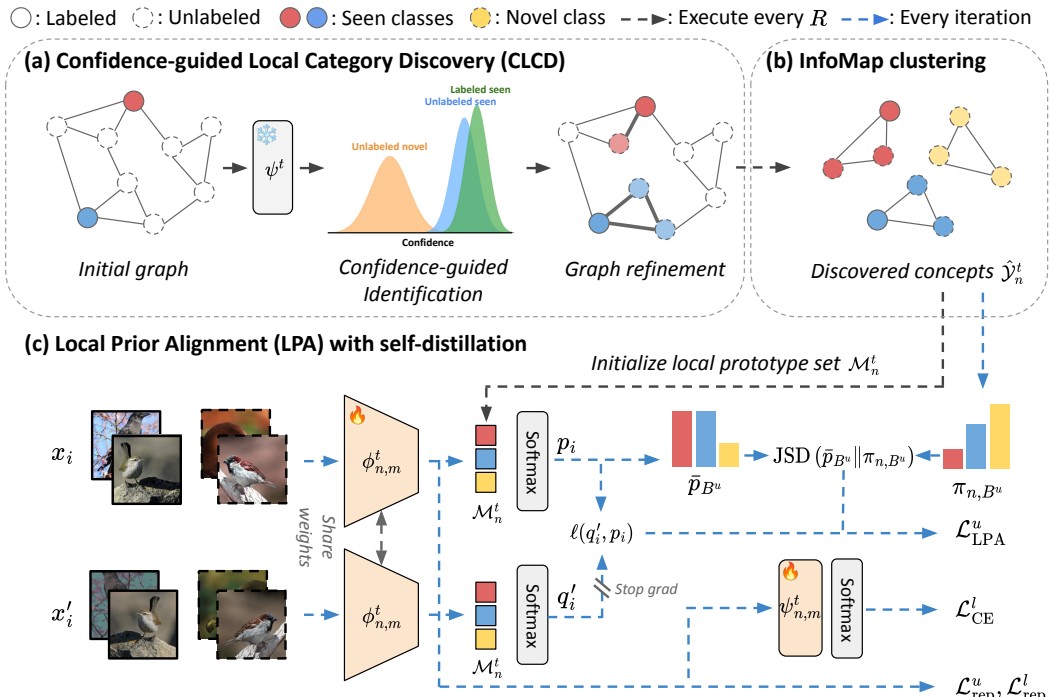

Figure 1: Overview of FedLPA's local training process. (a) Every $R$ rounds, each client constructs a local similarity graph using the current global backbone $\phi^t$ and refines it with (pseudo-)labels from seen classes. (b) Each client applies graph-based clustering (Infomap) on the refined graph, and obtains a local category prior $\hat{\mathcal{Y}}_n^t$ and corresponding concept prototypes $\mathcal{M}_n^t$. (c) During local training on unlabeled data, self-distillation minimizes cross-entropy between predictions $p_i$ and soft targets $q_i'$ derived from prototype similarities in $\mathcal{M}_n^t$. Concurrently, Local Prior Alignment (LPA) aligns the batch-averaged predictions $\bar{p}_{\mathcal{B}_u}$ with the batch-specific empirical class prior $\pi_{n,\mathcal{B}_u}$ via Jensen-Shannon Divergence (JSD). For labeled data, the framework employs standard cross-entropy and supervised contrastive loss, while unlabeled data additionally uses unsupervised contrastive loss.

## 3.2 Overview

FedLPA achieves robust generalized category discovery under severe data heterogeneity and class imbalance without requiring prior knowledge of novel class counts or imposing class-size balance assumption. As illustrated in Figure 1, we accomplish this through three synergistic stages at each client: (1) constructing an enriched local similarity graph via Confidence-guided Local Category Discovery (CLCD), (2) applying Infomap clustering to derive client-specific class priors, and (3) employing Local Prior Alignment (LPA) to align model predictions with these discovered local priors. This integrated approach enables robust category discovery across heterogeneous clients by grounding learning in each client's true data structure.

FedLPA follows the standard federated learning framework, FedAvg [15]. Specifically, a central server initializes a global model parameterized by $\theta = \{\phi, \psi\}$, comprising a feature extractor $f(\cdot; \phi)$ and a classifier $g(\cdot; \psi)$ dedicated to seen classes $\mathcal{Y}^l$. At each communication round $t \in \{1, \ldots, T\}$, the server distributes the global model $\theta^t$ to an active client subset $\mathcal{C}_t \subseteq \mathcal{C}$. Each client $C_n \in \mathcal{C}_t$ initializes its local parameters $\theta_{n,0}^t$ to $\theta^t$, and performs $M$ local optimization iterations on its local data $\mathcal{D}_n$. The server then aggregates the resulting local models $\theta_{n,M}^t$ and updates the global model $\theta^{t+1}$ for the next round of training by averaging the local model parameters. This training process is repeated until the global model $\theta^t$ converges.

## 3.3 Confidence-guided local category discovery

**Initial similarity graph construction** Each client $n$ constructs an initial similarity graph $G_n = \{\mathcal{I}_n, \mathcal{E}_n\}$ encoding pairwise feature relationships among its local samples $\mathcal{D}_n$. The node set $\mathcal{I}_n$

comprises all local samples $x_i \in \mathcal{D}_n$. Each edge weight $e_{ij} \in \mathcal{E}_n$ is the cosine similarity between $\ell_2$-normalized feature embeddings $v_i$ and $v_j$, extracted using the global backbone $f(\cdot; \phi^t)$ for samples $x_i$ and $x_j$, respectively. This initial graph $G_n$ establishes a foundational structure capturing pairwise feature affinities within the local data.

**Confidence-based known sample identification**   Since the initial graph relies solely on feature similarity, it is susceptible to noise and may not accurately reflect semantic relationships. To mitigate this, we refine the graph $G_n$ by incorporating supervisory signals from known categories $\mathcal{Y}^l$, identifying high-confidence pseudo-labels within the unlabeled set $\mathcal{D}_n^u$. Specifically, we utilize the global model $\theta^t = \{\phi^t, \psi^t\}$ received at the start of round $t$ (i.e., $\theta_{n,0}^t$) to compute logits $h(x; \theta^t) = g(f(x; \phi^t); \psi^t)$ over the seen classes $\mathcal{Y}^l$. For each unlabeled sample $x_i$, we define a confidence score $s(x_i)$ as the maximum softmax probability derived from these logits. If $s(x_i)$ exceeds a client-specific threshold $\xi_n$—determined as the $P$-th percentile of confidence scores on the local labeled data $\mathcal{D}_n^l$—we consider $x_i$ a reliably identified known sample. We assign a pseudo-label $\hat{y}_i \in \mathcal{Y}^l$ corresponding to the class with the highest probability, collecting these samples into a set:

$$\hat{\mathcal{D}}_n^{u,\text{seen}} = \{(x_i, \hat{y}_i) \mid x_i \in \mathcal{D}_n^u, \ s(x_i) > \xi_n, \text{ and } \hat{y}_i = \arg\max_{k \in \mathcal{Y}^l} \sigma(h(x_i; \theta^t))_k\}, \quad (1)$$

where $\sigma(\cdot)$ denotes the softmax function. To ensure reliable confidence estimates, particularly in early rounds, we employ an initial warm-up training phase. Further details on the warm-up procedure and the determination of $\xi_n$ are provided in the supplementary document.

**Label-informed graph refinement**   Leveraging the (pseudo-)labeled sample set, each client refines the edge weights $\mathcal{E}_n$ of its local similarity graph $G_n$ to explicitly encode these supervisory signals. Let $\mathcal{D}_n^{\text{sup}} = \mathcal{D}_n^l \cup \hat{\mathcal{D}}_n^{u,\text{seen}}$ be this set of samples with (pseudo-)labels $\tilde{y}_i \in \mathcal{Y}^l$. We recalibrate the initial edge weights $e_{ij} \in \mathcal{E}_n$ by imposing the following rules:

$$e'_{ij} \leftarrow \begin{cases} 1, & \text{if } x_i, x_j \in \mathcal{D}_n^{\text{sup}}, \tilde{y}_i = \tilde{y}_j, \text{ and } i \neq j \\ 0, & \text{if } x_i, x_j \in \mathcal{D}_n^{\text{sup}}, \tilde{y}_i \neq \tilde{y}_j, \text{ and } i \neq j \\ e_{ij}, & \text{otherwise} \end{cases} \quad (2)$$

To further enhance the graph's robustness against feature noise, we apply an edge-pruning mechanism. This step establishes the final edge set $\mathcal{E}'_n$ by filtering out edges with weights falling below a predefined threshold $\tau_f$. The resulting refined graph $G'_n = \{\mathcal{I}_n, \mathcal{E}'_n\}$ yields a more discriminative topology, thereby facilitating the subsequent local category discovery.

### 3.4  Infomap clustering

To uncover the latent semantic structure within the local data, each client applies the Infomap algorithm [21] to the refined graph $G'_n$. Infomap partitions the graph into communities by minimizing the description length of a random walk. This topological partitioning allows us to derive a set of concept assignments $\{c_i\}$ for each sample, thereby implicitly determining the number of discovered concepts $K_n = |\hat{\mathcal{Y}}_n^t|$.

Based on these assignments, the client constructs a set of $K_n$ local prototypes, denoted as $\mathcal{M}_n^t = \{\mu_{n,k}^t\}_{k=1}^{K_n}$. Specifically, each prototype $\mu_{n,k}^t$ is computed as the centroid of the $\ell_2$-normalized feature representations associated with concept $c_k$. These prototypes serve as semantic anchors for the self-distillation mechanism described in Section 3.5. To accommodate feature drift during training, this entire discovery and initialization process is periodically re-executed every $R$ communication rounds at the start of local training.

### 3.5  Local prior alignment (LPA) with self-distillation

Building upon the discovered local concepts, we introduce a robust self-distillation strategy augmented by a principled regularizer, termed Local Prior Alignment (LPA). During local training, the unsupervised objective for client $n$ on an unlabeled mini-batch $B^u \subset B$ is formulated as:

$$\mathcal{L}_{\text{LPA}}^u = \frac{1}{|B^u|} \sum_{x_i \in B^u} \ell(q'_i, p_i) + \varepsilon \, \text{JSD}(\bar{p}_{B^u} \,||\, \pi_{n,B^u}). \quad (3)$$

This objective combines an instance-level self-distillation loss $\ell(\cdot, \cdot)$ (cross-entropy) with the proposed LPA regularizer, balanced by the coefficient $\varepsilon$. The precise definitions and mechanisms of these two components are detailed below.

**Instance-level consistency**  The self-distillation component refines feature representations by enforcing predictive consistency across two augmented views, $x_i$ and $x_i'$, of the same image. For the first view $x_i$, the model computes a soft prediction vector $p_i$ based on the cosine similarity between its feature embedding $v_i = f(x_i; \phi_{n,m}^t)$ and the local prototypes $\mathcal{M}_n^t$. A sharpening operation with temperature $\tau_s$ is applied: $p_i \propto \exp(\text{sim}(v_i, \mu)/\tau_s)$. Simultaneously, a target distribution $q_i'$ is generated from the second view $x_i'$ using a lower temperature $\tau_t < \tau_s$ to produce a sharper supervision signal.

**Batch-level alignment**  The LPA regularizer acts as a distributional constraint, aligning the model's marginal prediction distribution with the inherent class distribution of the batch. We minimize the Jensen-Shannon Divergence (JSD) between the mean model prediction $\bar{p}_{B^u}$ and a batch-specific empirical class prior $\pi_{n,B^u}$. The prior distribution is derived from the frequency of pre-assigned concepts $\{c_j\}$ (obtained via Infomap) within the batch:

$$\pi_{n,B^u}[k] = \frac{1}{|B^u|} \sum_{x_j \in B^u} \mathbb{I}(c_j = c_k'), \quad k \in [K_n], \tag{4}$$

where $c_k'$ denotes the $k$-th concept in $\hat{\mathcal{Y}}_n^t$. The model's marginal distribution is computed by averaging the soft predictions from both views:

$$\bar{p}_{B^u} = \frac{1}{|B^u|} \sum_{x_i \in B^u} \frac{1}{2}(p_i + p_i'). \tag{5}$$

By enforcing this alignment, LPA adaptively guides the model towards the client's true local data structure and ensures robustness against severe class imbalance.

### 3.6  Joint optimization

To facilitate robust representation learning, we employ supervised [10] and self-supervised [4] contrastive losses, formulated as

$$\mathcal{L}_{\text{rep}}^l = \frac{1}{|B^l|} \sum_{i \in B^l} \frac{1}{|\mathcal{N}_i|} \sum_{q \in \mathcal{N}_i} -\log \frac{\exp\left(v_i^\top v_q'/\tau_c\right)}{\sum_i^{i \neq j} \exp\left(v_i^\top v_j'/\tau_c\right)}, \tag{6}$$

$$\mathcal{L}_{\text{rep}}^u = \frac{1}{|B|} \sum_{i \in B} -\log \frac{\exp(v_i^\top v_i'/\tau_u)}{\sum_i^{i \neq j} \exp(v_i^\top v_j'/\tau_u)}. \tag{7}$$

where $\mathcal{N}_i$ denotes all other samples in the batch with the same label as $x_i$, and $\tau_c, \tau_u$ are temperature parameters. For labeled data, we apply a standard cross-entropy loss $\mathcal{L}_{\text{CE}}^l$ on the labeled batch $B^l$ to train the local classifier $g(\cdot; \psi_{n,m}^t)$ for seen classes. Combining all components, the overall objective function for each client $n$ is:

$$\mathcal{L}_n = \lambda(\mathcal{L}_{\text{LPA}}^u + \mathcal{L}_{\text{rep}}^u) + (1-\lambda)(\mathcal{L}_{\text{rep}}^l + \mathcal{L}_{\text{CE}}^l), \tag{8}$$

where $\lambda$ is a hyperparameter balancing the loss terms. The detailed learning procedure of FedLPA is described in Algorithm 1 in the supplementary document.

## 4  Experiment

### 4.1  Experimental setup

**Dataset**  We evaluate our proposed method on six image classification benchmarks: three fine-grained datasets (CUB-200 [24], Stanford-Cars [11], and Oxford-IIIT Pet [17]) and three generic object recognition datasets (CIFAR-10 [12], CIFAR-100 [12], and ImageNet-100 [5]). For each dataset, we designate half of the classes as known and the other half as novel. From the known classes, 50% of instances comprise the labeled training subset, while the remaining instances, along

with all instances from novel classes, form the unlabeled training subset. To simulate non-*i.i.d.* data distributions across clients, we sample label proportions from a symmetric Dirichlet distribution with a concentration parameter $\alpha \in \{0.2, 0.05\}$, following [8, 19]. This procedure yields $|\mathcal{C}| = 5$ client-specific subsets, where each subset serves as a local dataset for an individual client.

**Baselines**   We compare our method, dubbed as *FedLPA*, with the state-of-the-art Fed-GCD methods: GCL [19], and AGCL [19]. We further establish federated baselines by adapting prominent centralized GCD methods—GCD [22], SimGCD [27], and GPC [30]—as well as an unsupervised learning method, PCL [13]. These methods are integrated with FedAvg [15] following the strategy in [19], and are denoted as *FedAvg + GCD*, *FedAvg + SimGCD*, *FedAvg + GPC*, and *FedAvg + PCL*, respectively. For *FedAvg + SimGCD*, we assume the number of novel classes is known a priori to initialize the classifier.

**Evaluation protocol**   We evaluate model performance using clustering accuracy (ACC) on an unlabeled test set held by the server, following the standard practice in [19]. This test set, along with a labeled validation set, is partitioned from a global evaluation set, mirroring the partitioning scheme used for the training data. Note that the baselines [19, 30, 22, 13] utilize the labeled validation data for either category number estimation or semi-supervised clustering. To ensure a direct and fair comparison with these baselines, we also report the performance of a variant, FedLPA+, which utilizes this validation set by applying our CLCD algorithm to guide semi-supervised clustering.

Given predicted labels $\hat{y}_i$ and ground-truth labels $y_i$, ACC is defined as follows:

$$ACC = \max_{\Pi \in S_k} \frac{1}{N_u} \sum_{i=1}^{N_u} \mathbf{1}\{\hat{y}_i = \Pi(y_i)\}, \tag{9}$$

where $S_k$ denotes the set of all possible permutations of $k$ cluster assignments, $N_u$ is the total number of unlabeled test samples, and $\Pi(\cdot)$ represents the optimal mapping found via the Hungarian algorithm. We report ACC for all unlabeled test samples ("All"), as well as separately for samples from "Old" classes ($y_i \in \mathcal{Y}^l$) and "New" classes ($y_i \in \mathcal{Y}^u \setminus \mathcal{Y}^l$).

**Implementation details**   We use a ViT-B/16 pretrained with DINO as the backbone. We use the output of the `[CLS]` token with a dimension of 768 as the feature for an image, and only fine-tune the last block of the backbone, following [19, 22]. The model undergoes a warmup stage of 20 rounds, followed by 50 rounds of Fed-GCD training. Both stages use SGD with a batch size of 128 and an initial learning rate of 0.1. For Fed-GCD training, the learning rate is decayed via a cosine schedule. Following [19], the number of local training epochs is set to 1 with full client participation. The balancing factor $\lambda$ is set to 0.35, the temperature values $\tau_s, \tau_c, \tau_u$ are set to 0.1, 0.07, 1.0, respectively. Following [27, 22], $\tau_t$ starts at 0.07 and anneals to 0.04 over the first 30 rounds using a cosine schedule. For FedLPA, we set the percentile $P$ to 80, the LPA regularization weight $\varepsilon$ to 0.5, and the CLCD update frequency $R$ as 1. We set $\tau_f$ to 0.6 and 0.4 for fine-grained and standard datasets, respectively. All experiments were conducted on a single NVIDIA RTX A6000 or A5000 GPU.

## 4.2   Results

We evaluate the proposed methods, FedLPA and FedLPA+, on six benchmarks, encompassing three fine-grained datasets and three standard object recognition datasets, under varying degrees of data heterogeneity. As detailed in Table 1 and Table 2, both FedLPA and FedLPA+ consistently outperform all existing Fed-GCD baselines across all datasets at every data heterogeneity level. Notably, FedLPA achieves these gains without any server-side labeled validation data, a common prerequisite for the baselines. This demonstrates FedLPA's robustness in realistic, resource-constrained federated settings, even when the server-side labeled validation data is not available. For direct comparison, FedLPA+ leverages the server-held validation data by applying our CLCD algorithm to the combined validation and unlabeled test sets, yielding further performance improvements in most cases. Among the baselines, FedAvg + SimGCD often struggles—particularly on standard object recognition datasets—likely due to its restrictive uniform-prior assumption. Similarly, FedAvg + GPC underperforms FedAvg + PCL in most cases, as it enforces balanced cluster size. These observations indicate that assuming class or cluster balance is ill-suited for the non-*i.i.d.* and imbalanced Fed-GCD settings, highlighting the advantages of our adaptive, data-driven structure discovery mechanisms.

Table 1: Results on fine-grained datasets with two different degrees of data heterogeneity. Methods with a dagger † report results from [19]. The 'Server val.' column indicates whether server-side labeled validation data is used for evaluation. Bold black and plain red numbers indicate the best and second-best performance, respectively, in each column.

| Method | Server val. | CUB-200 | | | | | | Stanford-Cars | | | | | | Oxford-Pet | | | | | |
|---|---|---|---|---|---|---|---|---|---|---|---|---|---|---|---|---|---|---|---|
| | | α = 0.2 | | | α = 0.05 | | | α = 0.2 | | | α = 0.05 | | | α = 0.2 | | | α = 0.05 | | |
| | | All | Old | New | All | Old | New | All | Old | New | All | Old | New | All | Old | New | All | Old | New |
| FedAvg + GCD† [22] | ✓ | 46.3 | 54.8 | 40.1 | 43.3 | 52.8 | 38.9 | 32.4 | 49.8 | 28.3 | 30.4 | 46.1 | 26.5 | 76.2 | 77.8 | 75.2 | 72.1 | 76.4 | 71.5 |
| FedAvg + SimGCD [27] | ✓ | 36.8 | 49.7 | 30.4 | 34.6 | 48.5 | 27.7 | 35.1 | 56.3 | 24.9 | 30.3 | 43.9 | 23.7 | 43.6 | 39.7 | 45.6 | 36.7 | 34.1 | 38.1 |
| FedAvg + PCL† [13] | ✓ | 51.3 | 53.5 | 49.8 | 47.5 | 53.0 | 46.3 | 35.3 | 47.7 | 33.4 | 32.6 | 45.5 | 29.2 | 79.4 | 80.3 | 79.1 | 76.6 | 77.9 | 74.7 |
| FedAvg + GPC† [30] | ✓ | 49.1 | 51.3 | 47.0 | 45.3 | 51.2 | 44.7 | 34.1 | 45.5 | 32.6 | 30.9 | 45.3 | 27.8 | 78.8 | 78.5 | 79.1 | 73.1 | 77.3 | 73.5 |
| FedAvg + GCL† [19] | ✓ | 53.7 | 54.6 | 53.2 | 52.2 | 53.1 | 52.9 | 36.0 | 48.1 | 33.7 | 35.3 | 45.7 | 31.5 | 80.7 | 81.3 | 80.2 | 79.5 | 81.5 | 78.6 |
| FedAvg + AGCL† [19] | ✓ | 55.2 | 52.5 | 56.7 | 53.1 | 52.9 | 54.2 | 38.2 | 50.8 | 36.0 | 36.4 | 44.9 | 32.8 | 82.7 | 83.9 | 82.3 | 81.4 | 82.0 | 80.7 |
| FedLPA (ours) | | 62.3 | 63.3 | 61.8 | 61.2 | 63.1 | 60.1 | 52.1 | 67.6 | 44.6 | 51.8 | 64.9 | 45.4 | 84.6 | 85.3 | 84.2 | 83.3 | 86.6 | 81.5 |
| FedLPA+ (ours) | ✓ | 63.5 | 63.6 | 63.4 | 62.6 | 64.3 | 61.8 | 57.7 | 70.1 | 51.7 | 54.2 | 69.4 | 46.8 | 86.7 | 90.7 | 84.7 | 85.0 | 86.9 | 84.0 |

Table 2: Results on standard object recognition datasets with two different degrees of data heterogeneity. Methods with a dagger † report results from [19]. The 'Server val.' column indicates whether server-side labeled validation data is used for evaluation. Bold black and plain red numbers indicate the best and second-best performance, respectively, in each column.

| Methods | Server val. | CIFAR-10 | | | | | | CIFAR-100 | | | | | | ImageNet-100 | | | | | |
|---|---|---|---|---|---|---|---|---|---|---|---|---|---|---|---|---|---|---|---|
| | | α = 0.2 | | | α = 0.05 | | | α = 0.2 | | | α = 0.05 | | | α = 0.2 | | | α = 0.05 | | |
| | | All | Old | New | All | Old | New | All | Old | New | All | Old | New | All | Old | New | All | Old | New |
| FedAvg + GCD† [22] | ✓ | 80.7 | 82.3 | 80.3 | 78.7 | 80.1 | 78.3 | 49.6 | 52.1 | 49.3 | 47.3 | 49.2 | 45.9 | 69.8 | 77.1 | 65.7 | 66.4 | 74.8 | 62.1 |
| FedAvg + SimGCD [27] | ✓ | 53.6 | 53.5 | 53.6 | 52.9 | 66.2 | 46.3 | 43.1 | 57.0 | 36.2 | 33.6 | 40.1 | 30.4 | 54.8 | 77.1 | 43.5 | 43.4 | 62.2 | 33.9 |
| FedAvg + PCL† [13] | ✓ | 81.6 | 82.7 | 80.9 | 80.0 | 80.7 | 79.4 | 53.2 | 54.1 | 51.7 | 50.4 | 51.6 | 49.0 | 72.4 | 79.5 | 66.0 | 70.1 | 77.0 | 63.3 |
| FedAvg + GPC† [30] | ✓ | 81.3 | 81.7 | 80.5 | 80.1 | 80.4 | 78.4 | 52.8 | 53.5 | 51.4 | 50.0 | 51.3 | 48.9 | 72.1 | 78.2 | 65.7 | 69.8 | 76.8 | 63.1 |
| FedAvg + GCL† [19] | ✓ | 83.2 | 84.9 | 82.8 | 82.2 | 82.4 | 81.9 | 54.1 | 55.7 | 54.0 | 52.1 | 53.2 | 51.9 | 74.1 | 81.8 | 67.3 | 72.5 | 79.8 | 65.3 |
| FedAvg + AGCL† [19] | ✓ | 84.7 | 85.5 | 84.6 | 82.5 | 83.4 | 82.2 | 56.1 | 56.8 | 55.3 | 54.2 | 54.6 | 54.0 | 74.8 | 80.2 | 69.8 | 73.1 | 78.1 | 67.0 |
| FedLPA (ours) | | 94.5 | 94.5 | 94.6 | 93.9 | 94.5 | 93.7 | 57.7 | 61.7 | 55.7 | 54.2 | 60.7 | 50.9 | 75.9 | 89.7 | 69.0 | 73.2 | 87.6 | 65.9 |
| FedLPA+ (ours) | ✓ | 95.1 | 96.3 | 93.9 | 94.1 | 95.1 | 93.3 | 58.1 | 63.4 | 55.4 | 56.5 | 64.9 | 52.3 | 76.6 | 90.6 | 69.9 | 74.4 | 88.6 | 67.3 |

## 4.3 Analysis

**Ablation study** To validate the efficacy of individual components within our FedLPA framework, we conduct an ablation study on Stanford-Cars under non-*i.i.d.* settings ($\alpha = 0.2$ and $\alpha = 0.05$), and the results are presented in Table 3. The results show that each proposed element contributes significantly to the final performance. Specifically, applying our Local Prior Alignment loss ($\mathcal{L}_{LPA}^{u}$) substantially enhances performance, even with a fixed target prior derived once from an initial graph of local unlabeled data (row 1). Our proposed regularizer on adaptive prior, empirically computed from each local batch, yields further significant gains (row 2). Notably, even without the confidence-guided graph refinement, our framework, LPA loss with an adaptive prior (row 3 and row 4), already demonstrates strong performance, outperforming all compared algorithms in Table 1. This highlights the robustness of our core LPA mechanism. Furthermore, our proposed confidence-guided graph refinement provides an additional performance gain (row 5).

**Analysis of CLCD algorithm** To validate the effectiveness of our proposed CLCD algorithm, we ablate the clustering module by replacing it with semi-supervised K-Means (used in [22, 30]) and semi-supervised FINCH (used in [19]) within the FedLPA framework, and the results are presented in Figure 2. It is important to note that for the implementation of both semi-supervised K-Means and FINCH, we assume that the knowledge of the true number of classes in both local data and the global test data is given. We report two outcomes: (a) clustering accuracy on each client's local data immediately after warmup (Figure 2a); and (b) final clustering accuracy on the server test set after 70 federated training rounds (Figure 2b). Our proposed method achieves superior clustering performance on local training data compared to the other algorithms. Consequently, this leads to a more significant improvement in the final test accuracy of FedLPA both on seen and novel classes, underscoring the effectiveness of the proposed CLCD algorithm on the overall model performance.

Table 3: Component analysis of the proposed methods in the non-*i.i.d.* settings on Stanford-Cars with two different degrees of data heterogeneity ($\alpha = 0.2$ and $\alpha = 0.05$). Bold black and plain red numbers indicate the best and second-best performance, respectively, in each column.

| $\mathcal{L}^u_{\mathrm{LPA}}$ | Target prior | Initial graph | CLCD | $\alpha = 0.2$ All | Old | New | $\alpha = 0.05$ All | Old | New |
|---|---|---|---|---|---|---|---|---|---|
| - | - | - | - | 34.1 | 50.8 | 26.0 | 32.0 | 50.3 | 23.1 |
| ✓ | Fixed | $\mathcal{D}^u_n$ | - | 48.3 | 61.7 | 41.8 | 47.3 | 55.5 | 43.2 |
| ✓ | Adaptive | $\mathcal{D}^u_n$ | - | 51.4 | 66.1 | 44.3 | 50.5 | 61.6 | 45.1 |
| ✓ | Adaptive | $\mathcal{D}^l_n \cup \mathcal{D}^u_n$ | - | *54.4* | *67.9* | *47.2* | *52.3* | *66.4* | *45.4* |
| ✓ | Adaptive | $\mathcal{D}^l_n \cup \mathcal{D}^u_n$ | ✓ | **57.7** | **70.1** | **51.7** | **54.8** | **69.4** | **46.8** |

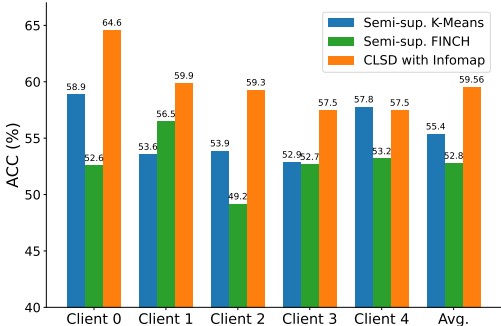

(a) Clustering accuracy on local data after warm-up

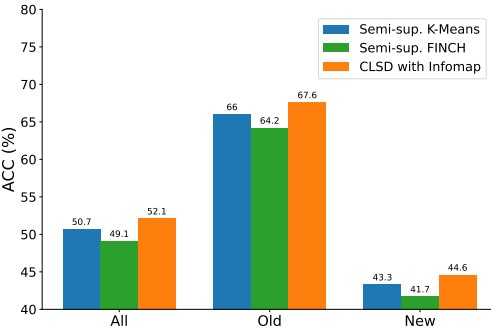

(b) Final clustering accuracy on server test data

Figure 2: Ablative results of CLCD algorithm in FedLPA under non-*i.i.d.* clients ($\alpha = 0.2$) on Stanford-Cars. We evaluate (a) clustering accuracy on individual client local training data right after the warmup training rounds, and (b) final clustering accuracy on the server test set after 70 federated training rounds. For the final clustering accuracy, all methods are evaluated identically at test time: we apply the same server-side Infomap clustering to the unlabeled test set with a fixed pruning threshold ($\tau_f = 0.6$), regardless of the clustering algorithm used during local training.

**Increased number of clients** We validate our framework in more challenging scenarios with an increased number of distributed clients ($N = 10$). All methods suffer from performance degradation, compared with the results in Table 4, due to the reduced local data per client, and increased data disparity. Despite these challenges, FedLPA consistently shows promising performance on all tested datasets.

**Hyperparameters** We investigate the impact of our hyperparameters on the performance of FedLPA under a non-*i.i.d.* setting with $\alpha = 0.2$, and the results are presented in Figure 3. Both the CLCD identification percentile $P$ (Figure 3a) and the Local Prior Alignment (LPA) regularization weight $\varepsilon$ (Figure 3b) demonstrate robust performance across a reasonable range of values, indicating FedLPA's stability. For the CLCD update frequency $R$ (Figure 3c), while more frequent updates ($R = 1$) yield better results by enabling rapid adaptation, FedLPA maintains competitive accuracy even with sparser updates. This offers a valuable trade-off, allowing for reduced computational overhead with only a marginal performance decrease, beneficial in resource-constrained federated scenarios. For the number of warmup rounds (Figure 3d), while a marginal performance drop is observed with very few initial rounds, FedLPA rapidly achieves competitive accuracy with a modest number of rounds (e.g., 10-20). The performance generally exhibits an upward trend and stabilizes as the number of warmup rounds increases (e.g., up to 50 rounds), indicating that sufficient warmup is beneficial.

## 5 Conclusion

We present Federated Local Prior Alignment (FedLPA), a novel framework for generalized category discovery in heterogeneous federated environments. Unlike prior approaches that rely on unrealistic global knowledge or fixed class priors ill-suited for federated settings, FedLPA operates entirely at the client level. The framework first constructs a client-specific similarity graph, enhanced by

Table 4: Results with increased number of clients ($N = 10$) on standard benchmarks in non-*i.i.d.* setting ($\alpha = 0.05$). Methods with a dagger † report results from [19]. The 'Server val.' column indicates whether server-side labeled validation data is used for evaluation. Bold black and plain red numbers indicate the best and second-best performance, respectively, in each column.

| Method | Server val. | CIFAR-10 | | | CIFAR-100 | | | ImageNet-100 | | |
|---|---|---|---|---|---|---|---|---|---|---|
| | | All | Old | New | All | Old | New | All | Old | New |
| FedAvg + GCD† [22] | ✓ | 63.4 | 60.0 | 66.7 | 47.3 | 48.3 | 45.6 | 62.3 | 70.8 | 60.1 |
| FedAvg + GCL† [19] | ✓ | 68.2 | 64.2 | 70.1 | 52.5 | 53.9 | 51.0 | 67.3 | 74.5 | 60.8 |
| FedAvg + AGCL† [19] | ✓ | 68.1 | 63.8 | 70.3 | 52.2 | 53.6 | 52.4 | 67.5 | 74.8 | 61.1 |
| FedLPA (ours) | | 92.3 | 94.5 | 91.2 | 53.6 | 54.5 | 53.1 | 71.7 | 85.6 | **64.7** |
| FedLPA+ (ours) | ✓ | **93.7** | **96.2** | **92.3** | **55.9** | **59.7** | **54.0** | **72.1** | **86.8** | 64.7 |

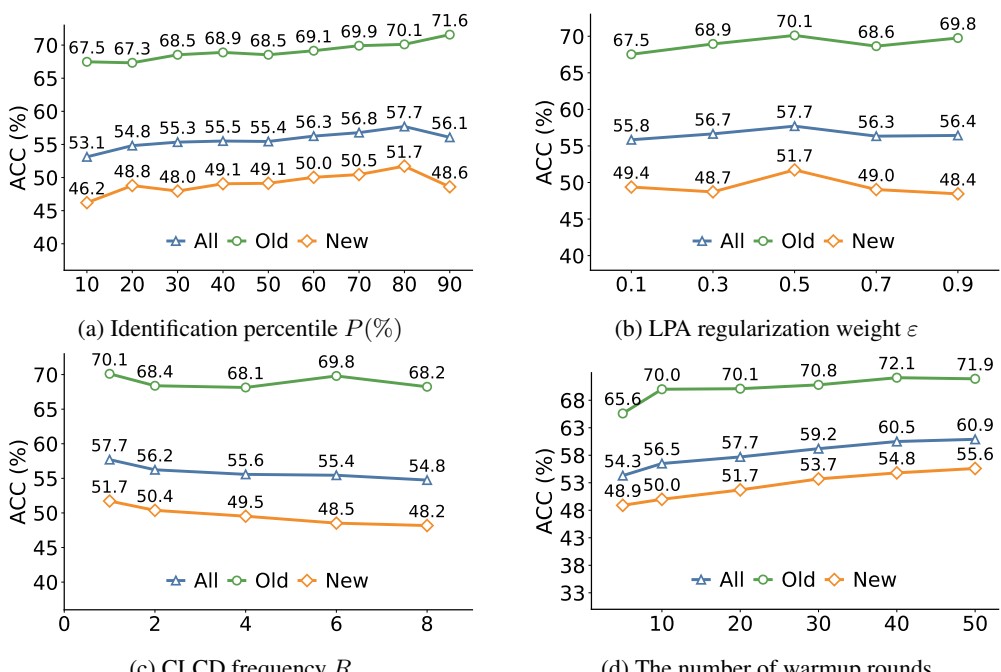

Figure 3: Ablative results of FedLPA hyperparameters in non-*i.i.d.* clients ($\alpha = 0.2$) on Stanford-Cars. We examine the impact of: (a) the percentile $P$ for known sample filtering in CLCD; (b) the weight $\varepsilon$ for the LPA regularizer in Eq (3); (c) communication rounds $R$ between CLCD executions; and (d) the number of rounds for federated warmup training.

reliably pseudo-labeled known-class samples, to capture local data structures without requiring global information or predefined category counts. Building on this foundation, our Local Prior Alignment (LPA) regularizer, integrated within a self-distillation scheme, dynamically adapts to local data distributions by aligning model predictions with an empirical class prior derived from these discovered structures. This synergy between local structure discovery and dynamic prior adaptation enables robust category discovery under severe data heterogeneity and class imbalance, yielding substantial performance gains over existing Fed-GCD methods across diverse benchmarks.

## Acknowledgements

This work was partly supported by the National Research Foundation of Korea (NRF) grant [RS-2022-NR070855, Trustworthy Artificial Intelligence] and by the Institute of Information & communications Technology Planning & Evaluation (IITP) grants [No.RS-2022-II220959 (No.2022-0-00959), (Part 2) Few-Shot Learning of Causal Inference in Vision and Language for Decision Making, No.RS-2025-25442338, AI star Fellowship Support Program(Seoul National University), No.RS-2021-II211343,

Artificial Intelligence Graduate School Program (Seoul National University), No.RS-2021-II212068, Artificial Intelligence Innovation Hub] funded by the Korean government (MSIT).

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
