# OpenReview forum: "FedLPA: Local Prior Alignment for Heterogeneous Federated Generalized Category Discovery"
_NeurIPS.cc/2025/Conference — NeurIPS 2025 poster_

### Official Review · Reviewer_QDuW · 2025-06-30

**Clarity:** 2
**Significance:** 2
**Originality:** 2
**Rating:** 5
**Confidence:** 4

**Summary:**

The paper proposes FedLPA, a framework for federated generalized category discovery (Fed-GCD) that avoids common unrealistic assumptions such as known novel class counts or uniform class distributions. The idea is to build client-specific similarity graphs enriched with pseudo-labels, apply InfoMap clustering, and align local model predictions with discovered local priors using a self-distillation-based regularizer. Experiments on several benchmarks under non-IID conditions are conducted to evaluate the effectiveness of the proposed method.

**Questions:**

1. The term "Local Prior Alignment" and use of "prior" throughout the paper give the impression of a Bayesian framework, but the method uses an empirical cluster distribution over batches (i.e., a histogram-based estimate). Can the authors clarify why they use the term "prior" in a non-Bayesian setting?
2. The paper lacks theoretical insights or guarantees, particularly concerning the stability and convergence of the self-distillation and alignment mechanisms across heterogeneous clients. Can the authors provide any theoretical or empirical analysis (even simplified) on the convergence behavior or alignment stability across clients? Have you observed cases where the learned cluster structures diverge across rounds?
3. Although the method avoids sharing raw data or prototypes, communication efficiency and privacy implications are largely unaddressed. What is the communication cost per round compared to FedAvg and AGCL? Could any of the learned local graph structures or soft predictions leak sensitive information? Have you considered adding differential privacy or gradient compression techniques?
4. The similarity graph used for InfoMap clustering is derived from features produced by an under-trained model early in training, which could lead to noisy clusters and unstable learning. Do you have quantitative or qualitative results showing how graph quality evolves across rounds? Have you considered alternatives like initializing from a pre-trained encoder (other than DINO), or delaying the clustering process until later rounds?

**Ethical Concerns:**

["NO or VERY MINOR ethics concerns only"]

**Final Justification:**

The authors addressed most of my concerns except the theoretical justification. I raise my score.

**Limitations:**

Yes

**Quality:**

2

**Strengths And Weaknesses:**

Strength:
Fed-GCD is underexplored yet important for real-world federated learning scenarios.

Weaknesses:
1. No theoretical analysis is provided regarding why the proposed local prior alignment scheme should converge or be stable across heterogeneous clients.
2. The model still relies on exchanging model parameters frequently. There’s limited discussion on communication cost or privacy guarantees (e.g., can prototype leakage reveal sensitive information?).
3. The title includes "Local Prior Alignment," which sounds Bayesian, but no formal Bayesian formulation or interpretation is provided.

---

> ### Author Rebuttal · Authors · 2025-07-31
>
> We truly appreciate your positive and constructive comments, and we will reflect on your feedback thoroughly. Here are the responses to the questions.
>
> **Q1. Can the authors provide any theoretical or empirical analysis on the convergence behavior?**
>
> **A1.** Thank you for your constructive feedback. While a formal theoretical convergence proof is challenging due to the non-differentiable nature of the InfoMap, we provide empirical evidence demonstrating the stable and efficient convergence of our method. Across all our experiments, the model consistently converges within the reported communication rounds (20 warm-up, 50 training). Note that FedLPA shows faster convergence compared to baselines like AGCL, which require up to 200 rounds for training. To provide a picture of convergence, we conducted an extended training run of 100 rounds (including warmup rounds). The results are presented in Table C below. As shown, both on CIFAR-100 and Stanford Cars, the performance converges around 60 rounds and maintains it thereafter. We will include the full convergence curves for all experiments in Tables 1 and 2 in the final manuscript.
>
> **Table C: Convergence of FedLPA on Stanford Cars and CIFAR-100 ($\alpha=0.2$). The table shows the "All" accuracy (%) over communication rounds.**
> |        Rounds      | 0    | 10    | 20   | 30    | 40   | 50   | 60   | 70   | 80    | 90   | 100  |
> | :----------- | :--: | :---: | :--: | :---: | :--: | :--: | :--: | :--: | :---: | :--: | :--: |
> | **Stanford-Cars** | 17.8 | 29.2 | 31.7 | 39.9 | 49.3 | 51.9 | 52.4 | 52.1 | 53.9 | 53.3 | 53.2|
> | **CIFAR-100** | 38.7 | 42.3 | 45.8 | 55.1 | 55.4 | 56.3 | 57.1 | 57.7 | 57.6 | 57.2 | 57.3|
>
>
>
>
>
>
> **Q2. Have you observed cases where the learned cluster structures diverge across rounds?**
>
> **A2.**  In our experiments, we found the learned cluster structures to be stably evolved and did not observe divergence during the learning process. We attribute this stability largely to the design of our CLSD mechanism. By establishing stable anchors within the similarity graph—using both ground-truth labels and high-confidence pseudo-labels from the seen classes—CLSD builds a reliable foundation for the graph. We believe this robust foundation mitigates drastic fluctuations in the overall cluster structure between rounds, providing a consistent learning target for the model.
>
> **Q3. Communication efficiency of FedLPA**
>
> **A3.** FedLPA is communication-efficient because it requires fewer total communication rounds to converge without incurring additional per-round costs compared to FedAvg.
>
> Since FedLPA only communicates model parameters, its per-round cost is identical to standard FedAvg and lower than methods like AGCL [19], which communicate extra information such as class prototypes. Furthermore, we empirically observed that FedLPA consistently converges within approximately 70 total rounds (20 warm-up and 50 training) across all datasets. This rapid convergence results in higher overall communication efficiency, as FedLPA reaches its final performance in fewer rounds than baselines requiring more extensive communication rounds (200 rounds reported in [19]).
>
>
> **Q4. Could any of the learned local graph structures or soft predictions leak sensitive information?**
>
> **A4.** No, FedLPA does not introduce additional privacy risks compared to the standard FedAvg protocol since FedLPA only communicates model parameters. Unlike methods such as AGCL that communicate class prototypes and risk leaking information about local class distributions, in FedLPA, the learned local graph structures, discovered concepts, and predictions are strictly confined to each client and never shared with the server or other clients. Therefore, FedLPA maintains the same privacy guarantees as FedAvg.
>
>
> **Q5. Have you considered adding differential privacy or gradient compression techniques?**
>
> **A5.** Thank you for the great suggestion. These techniques are orthogonal to our core method and can be readily integrated to enhance privacy and efficiency. We view this as a valuable direction for future work.
>
> **Q6. Do you have quantitative or qualitative results showing how graph quality evolves across rounds?**
>
> **A6.** We thank the reviewer for this constructive suggestion. We provide a quantitative analysis below, which shows that FedLPA consistently improves the quality of the graph-based local structures as training progresses.
> Table D presents the clustering accuracy on each client's local unlabeled data, which directly reflects the graph quality. The average accuracy steadily increases from 57.0% to 71.3%, demonstrating a positive feedback loop: better representations lead to higher-quality graphs, which in turn improve the model. This confirms that our framework reliably refines its understanding of local data structures throughout training.
>
> **Table D: Per-Client Clustering Accuracy (%) Progression of FedLPA on Stanford Cars ($\alpha = 0.2$)**
>
> | Rounds | 20 | 30 | 40 | 50 | 60 | 70 |
> | :------- | :------: | :------: | :------: | :------: | :------: | :------: |
> | Client 0 |   58.6   |   66.9   |   71.3   |   72.5   |   73.3   |   73.3   |
> | Client 1 |   55.6   |   62.4   |   65.5   |   67.4   |   67.9   |   69.2   |
> | Client 2 |   58.5   |   65.7   |   69.4   |   70.2   |   71.2   |   70.8   |
> | Client 3 |   57.8   |   69.2   |   72.1   |   73.1   |   74.4   |   74.0   |
> | Client 4 |   54.6   |   65.8   |   68.5   |   68.5   |   68.9   |   69.2   |
> | **Avg**  | **57.0** | **66.0** | **69.4** | **70.3** | **71.1** | **71.3** |
>
>
> **Q7. Have you considered alternatives like initializing from a pre-trained encoder (other than DINO), or delaying the clustering process until later rounds?**
>
> **A7.** Our framework is indeed flexible regarding the choice of the backbone; we used DINO primarily for a fair comparison with baselines.  To further validate your suggestion, we ran additional experiments on Stanford-Cars using a stronger DINOv2 backbone. The results show that FedLPA achieves a significant performance gain (e.g., ACC increased from 52.1% to 62.4% for $\alpha=0.2$ and from 51.8% to 58.1% for $\alpha=0.05$ ), highlighting that FedLPA directly benefits from the better backbone. We will include this result in the final manuscript. Thank you for this valuable suggestion.
>
> Regarding delaying the clustering, we have already explored this through our ablation study on the number of warm-up rounds (Fig. 4a in the supplementary document). As you suggested, the results show that performance improves with more warm-up rounds. Notably, our method achieves strong performance that already outperforms most baselines with as few as 10 warm-up rounds, demonstrating the robustness and efficiency of our approach.
>
> **Q8. The title includes "Local Prior Alignment," which sounds Bayesian, but no formal Bayesian formulation or interpretation is provided.**
>
> **A8.** To clarify, our method is not Bayesian. We use "Local Prior" to denote the empirical cluster distribution estimated from local data, which serves as a dynamic target for alignment, distinct from a Bayesian prior probability.

---

> > ### Author Response · Authors · 2025-08-03
> > **Look Forward to Your Feedback**
> >
> > Dear reviewer QDuW,
> >
> > Thank you again for your time and effort in reviewing our paper! As the discussion period is ending soon (August 6th), we would like to know if our responses have addressed your questions. If you have any remaining questions, please let us know so we can address them.
> > If our rebuttal has successfully addressed your concerns, we would be very grateful if you would consider reflecting this in your final evaluation. Thank you very much.

---

> > > ### Author Response · Authors · 2025-08-05
> > > **Waiting for Post-Rebuttal Feedback**
> > >
> > > Dear reviewer QDuW,
> > >
> > >
> > > We greatly appreciate your constructive comments, which are greatly helpful in strengthening our manuscript. In our rebuttal, we have thoroughly addressed all the raised concerns regarding the convergence, communication efficiency, potential privacy leakage, and stability of the proposed method.
> > >
> > >
> > > We remain available to clarify any further points or questions you might have. If our rebuttal has sufficiently addressed these concerns, we would be grateful if you would take our clarifications into account in your final assessment.
> > >
> > >
> > > We look forward to your feedback.
> > >
> > >
> > > Thank you,
> > >
> > > The Authors

---

> > ### Comment · Reviewer_QDuW · 2025-08-06
> >
> > Thank you to the authors for the rebuttal. While they have satisfactorily addressed the majority of my points, their answer to the theoretical analysis still does not fully convince me.

---

> ### Author Response · Authors · 2025-08-06
> **Further response to reviewer QDuW**
>
> Dear Reviewer QDuW,
>
> Thank you for your valuable feedback. To address your remaining concern regarding the theoretical analysis, we wish to first reiterate that a formal convergence proof for FedLPA is challenging due to the non-differentiability of InfoMap clustering, which precludes the use of standard analytical tools that rely on end-to-end differentiability.
>
> However, we believe a path toward analysis could emerge under a reasonable assumption that the discovered cluster structures will converge as FedLPA training progresses, even as the underlying feature representations are continuously refined.
> This assumption is empirically grounded; we observe that the clustering accuracy in each client converges in our experiments (Table D in the rebuttal). Under this assumption, we can conceptually decouple the discrete clustering process from the continuous model optimization. This simplification would then allow for a convergence analysis within a more conventional federated learning framework. While it does not represent a rigorous proof, it may offer valuable insight into the convergence behavior of FedLPA. We will add this discussion to the final manuscript.

---

### Official Review · Reviewer_Q1AF · 2025-07-02

**Clarity:** 3
**Significance:** 2
**Originality:** 2
**Rating:** 4
**Confidence:** 3

**Summary:**

This article considers generalized category discovery (GCD) under heterogeneous federated learning settings and proposes a novel self-distillation strategy. The FedLPA framework leverages the synergy between category structures identified through clustering and dynamic prior adaptation, enabling robust generalized category discovery under severe data heterogeneity. Additionally, this proposed framework successfully overcomes these unrealistic assumptions by adaptively estimating the category structure at the client level, significantly enhancing the practical value. Extensive experiments further demonstrates its performance.

**Questions:**

pls refer to the weakness part.

**Ethical Concerns:**

["NO or VERY MINOR ethics concerns only"]

**Final Justification:**

My concerns have almost been addressed, so i will keep my positive score.

**Limitations:**

yes

**Quality:**

3

**Strengths And Weaknesses:**

Strengths.
1. The paper is technically sound, with a well-motivated and novel approach.
2.  The paper is well-organized and clearly written.
3. This work addresses the critical challenges of generalized category discovery (GCD) in federated learning. By proposing a method that relaxes unrealistic assumptions common in prior work, it pushes the field towards more realistic application scenarios.
4. The core contribution—the Confidence-guided Local Structure Discovery (CLSD) pipeline and the Local Prior Alignment (LPA) regularizer is a novel idea for the Fed-GCD setting. Built upon existing concepts like graph-based clustering and self-distillation , the FedLPA framework is a well-crafted and original combination of techniques for this specific problem.

Weaknesses.

1.	The method requires each client to construct a similarity graph and run InfoMap clustering on all local data during communication rounds. You mentioned that more frequent updates (R = 1) yield better results. However, for clients with large amounts of data or limited computational resources, this could pose a significant computational bottleneck. The paper does not quantify this overhead or discuss potential optimization strategies.

2.	You have selected JSD (Jensen-Shannon Divergence) in the LPA regularizer term. This is a reasonable choice as it is symmetric and bounded. I wonder if you have tried other divergence measures (e.g., simple KL divergence)? Is there any particular theoretical or empirical reason for choosing JSD? More details would be helpful.

3.	CLSD relies on the classifier trained during the warm-up phase. Could you elaborate on how sensitive this method is to the quality of warm-up? For example, if the warm-up is shortened to fewer rounds, how much would the overall performance degrade? This would help in understanding the robustness of the framework.

---

> ### Author Rebuttal · Authors · 2025-07-31
>
> We sincerely thank the reviewer for the thoughtful feedback and for acknowledging the key strengths of our work. We are grateful for your positive comments on our model being technically sound, well-motivated, and addressing a critical research problem. We address your questions below.
>
>
> **Q1. The method requires each client to construct a similarity graph and run InfoMap clustering on all local data during communication rounds. You mentioned that more frequent updates (R = 1) yield better results. However, for clients with large amounts of data or limited computational resources, this could pose a significant computational bottleneck. The paper does not quantify this overhead or discuss potential optimization strategies.**
>
>
> **A1.** We thank the reviewer for this practical concern. While the local structure discovery (CLSD) phase indeed introduces additional computational cost, we believe that the resulting performance gains significantly outweigh this overhead. Crucially, this cost can also be flexibly managed.
>
> To quantify the cost, our experiments on fine-grained datasets (when $R=1$ with a ViT-B/16 backbone on a single A6000 GPU) show the CLSD process takes approximately 7.2 seconds per client, constituting about 11.4% of a 63.12-second wall-clock time for a local training round.
>
> More importantly, this overhead is not a fixed bottleneck. As shown in Figure 2c in the main paper, practitioners can directly control this cost by adjusting the CLSD frequency, $R$. The framework maintains highly robust accuracy even when CLSD is performed less frequently. For instance, by setting $R=8$, the CLSD component is executed only once every eight rounds. This reduces its overall contribution to the total computational cost from 11.4% to just 1.42%, while still maintaining competitive performance. This flexibility offers a practical way to adapt our method to various resource-constrained environments.
>
>
> **Q2. I wonder if you have tried other divergence measures (e.g., simple KL divergence)? Is there any particular theoretical or empirical reason for choosing JSD? More details would be helpful.**
>
>
> **A2.** Thank you for the excellent question. You are correct; we chose Jensen-Shannon Divergence (JSD) for our LPA regularizer precisely because its symmetric and bounded properties led to empirically stable training.
> We did experiment with the standard KL divergence, but we found the training to be unstable. The loss often became biased towards high-probability classes or would explode due to near-zero probabilities in the model's predictions. Even with heuristic adjustments such as adding an epsilon value or clipping the distribution, the instability persisted. In contrast, JSD provided a consistently robust and reliable training signal without such modifications.
>
>
> **Q3. How sensitive is this method to the quality of warm-up?**
>
>
> **A3.** We have analyzed the sensitivity to the warm-up quality through an ablation study on the number of warm-up rounds, and the results are illustrated in Fig. 4a in the supplementary document. The results show that while performance improves with more warm-up rounds, our method is not overly sensitive to it, which demonstrates the robustness and efficiency of our approach.

---

> > ### Author Response · Authors · 2025-08-03
> > **Look Forward to Your Feedback**
> >
> > Dear reviewer Q1AF,
> >
> > Thank you again for your time and effort in reviewing our paper! As the discussion period is ending soon (August 6th), we would like to know if our responses have addressed your questions. If you have any remaining questions, please let us know so we can address them.
> > If our rebuttal has successfully addressed your concerns, we would be very grateful if you would consider reflecting this in your final evaluation. Thank you very much.

---

> > > ### Author Response · Authors · 2025-08-05
> > > **Waiting for Post-Rebuttal Feedback**
> > >
> > > Dear reviewer Q1AF,
> > >
> > >
> > >
> > > We are grateful for your insightful comments, which are greatly helpful in improving the quality of our paper. In our rebuttal, we have carefully addressed your concerns regarding our method's computational overhead, divergence measure, and sensitivity to warm-up.
> > >
> > >
> > >
> > >
> > >
> > >
> > > If you have any remaining concerns, please do not hesitate to let us know, as we are ready to discuss them further. Otherwise, we would deeply appreciate it if you would consider our detailed response in your final evaluation.
> > >
> > >
> > >
> > > We look forward to your feedback.
> > >
> > >
> > > Best regards,
> > >
> > >
> > > The Authors

---

> > ### Comment · Reviewer_Q1AF · 2025-08-05
> >
> > Thank you for the response. My concerns have almost been addressed, so I will keep my score.

---

### Official Review · Reviewer_3rHf · 2025-07-02

**Clarity:** 3
**Significance:** 3
**Originality:** 3
**Rating:** 4
**Confidence:** 4

**Summary:**

This work tackles the challenging problem of generalized category discovery in federated learning (Fed-GCD). The main challenges mainly come from data heterogeneity (each client sees a different subset of classes with varying distributions), unknown and non-uniform category counts, and imbalanced class distributions. To address these challenges, the authors propose FedLPA, a client-side framework that discovers local category structures via graph-based clustering and enhances representation learning through a self-distillation strategy called Local Prior Alignment (LPA). By aligning model predictions with dynamically estimated local priors, FedLPA adapts to each client’s unique data distribution without requiring global class counts or balanced data. Experiments show that FedLPA significantly outperforms existing Fed-GCD approaches.

**Questions:**

1. Could global novel-class prototypes be gradually learned via communication of aligned clusters or prototypes?
2. How sensitive is FedLPA to noisy pseudo-labels or graph errors in the early training stage (when the quality is not good enough)?
3. How does FedLPA compare to pure federated contrastive learning methods? Since FedLPA does not optimize the global novel-class prototypes via federated learning and mainly focuses on the structure of the embedding space, it is worth comparing with the federated contrastive learning methods, which also mainly focus on the structure of the embedding space.

**Ethical Concerns:**

["NO or VERY MINOR ethics concerns only"]

**Final Justification:**

I confirm this review.

**Limitations:**

Yes.

**Quality:**

3

**Strengths And Weaknesses:**

Strengths:
1. This work removes some unrealistic assumptions (e.g., a known number of novel classes, balanced distributions) commonly used in the open-world learning research field, making the Fed-GCD problem setting more realistic.
2. The method contains two novel modules, CLSD and LPA, which guide self-supervised learning using local graph structure and batch-level priors.
3. Rich experiments show that the proposed method outperforms state-of-the-art Fed-GCD baselines on both fine-grained and generic datasets.

Weaknesses:
1. Local clusters are not aligned across clients. For example, the discovered novel class "A" on one client has no correspondence to "A" on another client, which can be problematic, especially for novel class discovery.
2. This method heavily relies on the quality of local clustering. However, in this federated learning setting, a few clients may encounter noisy features or samples. Under this condition, the quality of local clustering for some clients can be quite poor, which can negatively impact the performance of the global model.

---

> ### Author Rebuttal · Authors · 2025-07-31
>
> We appreciate the constructive and insightful feedback. We address your questions below and welcome any further questions.
>
>
> **Q1. Local clusters are not aligned across clients. For example, the discovered novel class "A" on one client has no correspondence to "A" on another client, which can be problematic, especially for novel class discovery. Could global novel-class prototypes be gradually learned via communication of aligned clusters or prototypes?**
>
>
> **A1.** We thank the reviewer for this insightful comment. FedLPA tackles the misalignment problem via relational modeling. Specifically, seen-class representations act as globally consistent anchors, learned with shared semantics across the federation. By modeling each client's novel classes in relation to these common anchors (via local relational graph construction), our approach can establish a consistent relational structure without requiring direct semantic alignment of the novel classes themselves. This, in turn, enables consistent representation learning for all classes by focusing on their relationships to the shared anchors.
>
> Our ablation study (Row 5 in Table 3) validates this anchor-based mechanism. The graph refinement establishes a clearer relational structure using seen-class examples, resulting in performance improvement.
>
>
> **Q2. This method heavily relies on the quality of local clustering. However, in this federated learning setting, a few clients may encounter noisy features or samples. Under this condition, the quality of local clustering for some clients can be quite poor, which can negatively impact the performance of the global model.**
>
> **A2.** We thank the reviewer for the constructive comments. To analyze the sensitivity of FedLPA to the quality of local clustering, we conduct an experiment on Stanford-Cars ($\alpha=0.2$) by intentionally injecting noise into the pseudo-labels generated by our Confidence-guided Local Structure Discovery (CLSD) module. At the beginning of every CLSD phase (every $R=5$ rounds), we randomly flipped a certain percentage of the pseudo-labels to other classes.
>
>
> The results, presented in Table B, show that while performance degrades gracefully as the label noise rate increases, FedLPA maintains robust performance. Even with a 30% noise rate, the accuracy remains competitive and outperforms the baselines reported in our main paper (Table 1). This demonstrates that our framework is not overly sensitive to a moderate level of noise in local clustering and can effectively mitigate its negative impact.
>
>
> **Table B: Robustness of FedLPA to noisy pseudo-labels on Stanford-Cars ($\alpha=0.2$). We report "All" accuracy (%) under varying rates of injected label noise.**
>
> | Pseudo-label Noise Rate | All | Old | New |
>  |------------|------|------|------|
> | 0 % |  52.1 | 67.6 | 44.6 |
> | 10 % |  48.3 | 60.1 | 42.6 |
>  |20 % |  47.7 | 63.0 | 40.3 |
> | 30 % |  47.2 | 59.7 | 41.1 |
>
>
> **Q3. How does FedLPA compare to pure federated contrastive learning methods?**
>
>
> **A3.** Direct comparison with the other federated contrastive learning methods, such as MOON [Li21], FedRCL [Seo24], and FedMoco [Dong21], is not straightforward since our work focuses on discovering novel classes in unlabeled data and learning proper representations along with seen classes, while these methods are not inherently designed to discover or model the underlying semantic category structures, which is the central challenge in Fed-GCD and the primary focus of FedLPA.
>
>
> **References**
>
> [Li21] Q. Li, et al., Model-Contrastive Federated Learning, in CVPR, 2021
>
> [Seo24] S. Seo, et al., Relaxed Contrastive Learning for Federated Learning, in CVPR, 2024
>
> [Dong21] N. Dong, et al., Federated Contrastive Learning for Decentralized Unlabeled Medical Images, in MICCAI, 2021

---

> > ### Comment · Reviewer_3rHf · 2025-08-01
> >
> > I appreciate the authors' efforts in presenting a detailed rebuttal, and hence I will keep my positive rating.

---

### Official Review · Reviewer_mcj9 · 2025-07-03

**Clarity:** 3
**Significance:** 3
**Originality:** 3
**Rating:** 4
**Confidence:** 5

**Summary:**

- The authors propose a federated learning (FL) algorithm for the Generalized Category Discovery (GCD) setting, dubbed as FedLPA, where partial data samples are unknown (i.e., the ground-truth label is not provided in the overall dataset) and unlabeled (i.e., the label of the sample is not given).
- The key part of FedLPA is the Local Prior Alignment (LPA) loss term, which combines two terms: a contrastive perspective term (comparing an unlabeled sample and its augmented one) and the distributional divergence term (comparing the computed logit distribution and its empirical prior).
- Through the extensive simulation results, FedLPA outperforms baselines with the extensions of the GCD methods to the FL setting. Also, the ablation studies have shown that the LPA term brings substantial performance gains.

**Questions:**

My main questions are based on Weaknesses 1 and 2.
- Weakness 1 is based on the concern that contributions are highly focused on the technical improvement, rather than showing the insights: why we use LPA, how LPA relieves the challenges of FL+GCD, and further impacts on other FL+GCD methods.
- Weakness 2 is based on the limited analysis of convergence, either in theoretical and empirical analysis.
- Please refer to the Weakness part.

Additional minor questions are as follows:
- I am not sure how to compute the empirical prior $\pi_{n, B_u}$ in the LPA term (Eq. 3). For unlabeled or unknown samples, we do not know the actual empirical distribution. Thus, I wonder how to compute the empirical prior. To the best of my understanding, it would be possible to compute the empirical prior of the known/labeled samples while leaving the prior of unlabeled or unknown samples vacant (or equally-distributed). However, I am not sure the exact process. Please elaborate on the computation of $\pi_{n, B_u}$.

**Ethical Concerns:**

["NO or VERY MINOR ethics concerns only"]

**Final Justification:**

After the review and rebuttal, the advantages of this work seem to outweigh the negatives. I believe that the paper meets the level of NeurIPS.

**Limitations:**

The authors have identified some limitations, such as adaptive client sampling and asynchronous updates, which are common challenges encountered in practical FL systems. These are convincing comments when focusing on the practicality of FL.
However, my additional comments on the limitations are based on the essential understanding of the proposed method, and in-depth analysis, as aforementioned in Weaknesses 1 and 2. Please refer to Weaknesses 1 and 2.

**Paper Formatting Concerns:**

I cannot find the paper formatting concerns.

**Quality:**

3

**Strengths And Weaknesses:**

**Strength 1: Convincing and timely extensions of the GCD problem to the FL setting**
- The main strength of this paper is demonstrating the technical improvement in the convincing and timely combinations between GCD and FL. In the future, it is anticipated that a deep model will undergo lifelong training in distributed agents, enabling it to discover novel concepts or categories. With this perspective, as done in this paper, the extensions of GCD to FL and the technical advances are meaningful steps in the related research society.

**Strength 2: Extensive simulation results and the substantial performance gains**
- The proposed method, i.e., FedLPA, has been extensively tested on various datasets, ranging from simple ones such as CIFAR and CUB-200 to more complex ones, including ImageNet, Stanford-Cars, and Oxford-Pet. I am well convinced that the proposed method is truly effective in the various FL+GCD scenarios.
- In addition, the performance gains are quite substantial beyond the baselines. For example, in CIFAR-10, the gains exceed +10%, and this trend is consistent across all cases. I believe that the gains are meaningful in realizing a practical FL method successfully working in the GCD scenario.

**Weakness 1: Limited insights into the broader impact of the FL+GCD topic**
- To my understanding, the key contributions of this paper are focused on technical improvements for FL+GCD. The authors have employed delicate techniques, including precise procedures for local structure discovery, clustering, and the combination of various loss terms (including the LPA term).
- From this perspective, it is difficult to identify the novelty of this work, particularly in envisioning how FL+GCD should be. When focusing on the key part of this work, i.e., LPA, the authors should provide concrete evidence: **i)** Why should we compare the logit distribution and the empirical distribution (referring to the second term in Eq. 3)?, **ii)** How does the term relieve the challenges in FL+GCD (when referring to the limitations of the existing approaches listed in the Introduction)?, and **iii)** Any broader impacts and applications of this loss term design to other FL+GCD approaches?
- To my understanding, other technical parts, including local graph structure discovery and InfoMap, can be viewed as borrowed and assembled parts from the well-known prior works (we know that graph construction and revision are widely used in transductive and semi-supervised approaches. For example, Tranductive Propagation Net is a method to find/revise the graph of novel samples, which is a clear example [TPN'18]). Thus, I hope to narrow my viewpoint to the LPA part when assessing the technical novelty.
- However, the technical design of LPA is shown, but the reasoning for this design, the role in relieving the challenges of FL+GCD, and possible extensions of LPA have not been addressed.

[TPN'18] Learning to Propagate Labels: Transductive Propagation Network for Few-shot Learning, ICLR'19

**Weakness 2: Limited discussion on convergence (both in the empirical and theoretical viewpoints)**
- In the FL society, it is crucial to provide a convergence guarantee for the newly proposed FL algorithm. However, in this paper, I cannot find any evidence of the FedLPA's convergence, either from the empirical or theoretical viewpoints.
- When a FedAvg is done with the LPA loss, is it possible to provide a theoretical convergence analysis? Also, if the theoretical analysis is too challenging, is it possible to provide the convergence behavior of FedLPA across rounds, `at least' via empirical results?
- For more comments on the empirical investigation, I hope to know how fast FedLPA converges across rounds. For example, does FedLPA show a reasonable convergence when compared to other baselines? I am quite worried that FedLPA shows a slow convergence due to the highly delicate design of various loss terms. It would be beneficial to see the required rounds to saturate for each algorithm in Table 1 and 2.

---

> ### Author Rebuttal · Authors · 2025-07-31
>
> We appreciate the constructive feedback. We provide the following responses to address the questions.
>
> ---------------------------
>
> **Q1. Novelty and broader import of FedLPA**
>
> We thank the reviewer for the constructive feedback. We will address the core concern about the novelty and broader impact of our work.
>
>
> Our primary contribution is a novel Fed-GCD framework that **removes the unrealistic assumptions** prevalent in prior GCD works, both centralized and federated settings. Previous GCD methods, including federated adaptations, are built on two unrealistic assumptions in a real-world federated environment:
>
>
> - **Global Knowledge Assumption:** Most GCD methods like SimGCD [27], GCD [22], and even FedoSSL [28] require a priori knowledge of the total number of novel classes. In a federated system with strict privacy, no single client or server can possess this global information.
>
> - **Uniform Distribution Assumption**: Many approaches [16, 20, 23, 25, 27, 28, 30] enforce a uniform class distribution prior (e.g., via entropy maximization), which doesn’t make sense considering the highly skewed and heterogeneous nature of FL.
>
>
> We propose a framework that effectively tackles the Fed-GCD problem by removing these assumptions. Specifically, our Confidence-guided Local Structure Discovery (CLSD) obviates the need for global knowledge by enabling each client to autonomously discover its unique local category counts and distributions. Building upon this, our Local Prior Alignment (LPA) then removes the uniform class distribution assumption by dynamically aligning the model's predictions with the actual local data distribution. CLSD and LPA mutually benefit each other, making our framework robust to severe data heterogeneity and fostering effective category discovery across clients.
>
> We now address the reviewer's specific questions about LPA and CLSD.
>
>
> **Q1.1. Why does LPA compare logit and empirical distributions, and how does the LPA term relieve the challenges in FL+GCD?**
>
> **A1.1.** LPA achieves robust regularization by aligning the model's prediction distribution with an empirical prior—a "softer" approach that is more resilient to noise than using hard pseudo-labels. Since individual cluster assignments from clustering can be noisy, using these as hard targets for cross-entropy loss or contrastive loss would make the model highly sensitive to this noise, forcing overconfidence in incorrect labels. Instead, by matching the model's average prediction distribution on a batch to the empirical cluster distribution discovered by CLSD, LPA guides the model’s overall behavior to reflect the client’s class distribution. This makes the learning process inherently more stable and resilient to noise from the clustering step.
>
>
> This directly resolves a major limitation of existing GCD works. Unlike previous methods that enforce uniform class distribution, LPA replaces this fixed, unrealistic assumption with a dynamic, data-driven regularization. It actively guides the model to learn each client's unique class distribution, thereby robustly handling the severe data heterogeneity and class imbalance that are unique challenges in Fed-GCD.
>
>
> **Q1.2. Technical novelty in local graph structure discovery and InfoMap**
>
>
> **A1.2.** While we utilize established tools like InfoMap, our technical novelty lies in their principled adaptation to solve a critical sub-problem in Fed-GCD: the assumption-free estimation of the unknown number and distribution of local categories at each client. Our proposed CLSD is a discovery engine whose primary function is to provide the dynamic, client-specific data distribution necessary for our LPA to work. This ability to perform robust local category discovery is a non-trivial prerequisite for the realistic Fed-GCD solution.
>
>
>
> **Q1.3. Any broader impacts and applications of this loss term design to other FL+GCD approaches?**
>
>
> **A1.3.** LPA can be extended beyond our specific framework, as its core principle offers a principled and robust alternative to the unrealistic uniform prior assumption. We believe LPA can be readily adapted as a "drop-in" module in existing parametric GCD methods (e.g., SimGCD), extending their applicability to the realistic, imbalanced data distributions where they struggle, in both centralized and federated settings.
>
> -------------------------------
>
> **Q2. When a FedAvg is done with the LPA loss, is it possible to provide a theoretical convergence analysis? Also, if the theoretical analysis is too challenging, is it possible to provide the convergence behavior of FedLPA across rounds, or via empirical results?**
>
>
> **A2.** Thank you for your constructive feedback. While a formal theoretical convergence proof is challenging due to the non-differentiable nature of the InfoMap, we provide empirical evidence demonstrating the stable and efficient convergence of our method. Across all our experiments, the model consistently converges within the reported communication rounds (20 warm-up, 50 training). This demonstrates faster convergence compared to baselines such as AGCL, which require up to 200 rounds for training. To provide a picture of convergence, we conducted an extended training run of 100 rounds (including warmup rounds). The results are presented in Table A below. As shown, both on CIFAR-100 and Stanford Cars, the performance converges around 60 rounds and maintains it thereafter. We will include the full convergence curves for all experiments in Tables 1 and 2 in the final manuscript.
>
>
> **Table A: Convergence of FedLPA on Stanford Cars and CIFAR-100 ($\alpha=0.2$). The table shows the "All" accuracy (%) over communication rounds.**
>
> |        Rounds      | 0    | 10    | 20   | 30    | 40   | 50   | 60   | 70   | 80    | 90   | 100  |
> | :----------- | :--: | :---: | :--: | :---: | :--: | :--: | :--: | :--: | :---: | :--: | :--: |
> | **Stanford-Cars** | 17.8 | 29.2 | 31.7 | 39.9 | 49.3 | 51.9 | 52.4 | 52.1 | 53.9 | 53.3 | 53.2|
> | **CIFAR-100** | 38.7 | 42.3 | 45.8 | 55.1 | 55.4 | 56.3 | 57.1 | 57.7 | 57.6 | 57.2 | 57.3|
>
> -------------------------------
>
> **Q3. How to compute the empirical prior?**
>
>
> **A3.** The empirical prior $\pi_{n, B_{u}}$ is not computed from ground-truth labels, but is instead empirically derived from the pseudo-labels generated by our framework. Specifically, at the start of each local training phase, our CLSD module and InfoMap (Sec. 3.3-3.4) assigns a cluster ID (a pseudo-label) to every local sample. Following Eq. 4 in the main paper, the prior $\pi_{n, B_{u}}$ is then simply the normalized histogram of these pre-assigned pseudo-labels for the samples within the current unlabeled mini-batch $B_{u}$.
>
>
> We hope our responses have addressed your concerns. We welcome any further questions.

---

> > ### Author Response · Authors · 2025-08-03
> > **Look Forward to Your Feedback**
> >
> > Dear reviewer mcj9,
> >
> > Thank you again for your time and effort in reviewing our paper! As the discussion period is ending soon (August 6th), we would like to know if our responses have addressed your questions. If you have any remaining questions, please let us know so we can address them.
> > If our rebuttal has successfully addressed your concerns, we would be very grateful if you would consider reflecting this in your final evaluation. Thank you very much.

---

> > > ### Author Response · Authors · 2025-08-05
> > > **Waiting for Post-Rebuttal Feedback**
> > >
> > > Dear reviewer mcj9,
> > >
> > > We thank you for taking the time to provide constructive comments, which are greatly helpful in improving the quality of our paper. We have thoroughly addressed all the raised concerns regarding the novelty and convergence of the proposed method.
> > >
> > >
> > > If you have any remaining concerns, then feel free to let us know. We would be happy to continue the discussion if there are additional questions.
> > > Otherwise, we would be very grateful if you would consider our response and clarifications in your final evaluation.
> > >
> > >
> > > We look forward to your feedback.
> > >
> > >
> > > Best regards,
> > >
> > > The Authors

---

> > ### Comment · Reviewer_mcj9 · 2025-08-06
> >
> > Dear authors,
> >
> > I truly appreciate your kind and detailed response to relieve my concerns.
> > First, I agree that your pipeline shows a considerable contribution to relieving the uniform-prior assumptions of the existing works. Thus, I understand why LPA, which compares the distributions, is required. In a nutshell, I conjecture that it is required to work on non-uniform distributions. I acknowledge that the proposed pipeline has clear potential to broadly improve GCD research, not confined within the FL scenario. I hope that it can be more emphasized in the revised paper, if accepted.
> >
> > It's impressive to hear that your pipeline shows a faster convergence than the baseline (approximately 2x faster?). At first glance, I was slightly worried that it would probably lead to a lagged convergence due to the complicated loss terms. Still, at this moment, considering the unique capability to work with non-uniform priors, I can accept the faster convergence behavior. I hope that it can be emphasized in the revised paper, if accepted.
> >
> > Finally, as a minor point, I believe that any FL should strive to provide mathematical guarantees for convergence (even with relatively strong assumptions). I also agree that the theoretical analysis is too challenging for the delicately designed methods. Still, theory should be pursued because FL is a decentralized optimization problem, where the convergence guarantee is the crucial factor. However, I think the advantages outweigh the minor point.
> >
> > Based on the addressed concerns and a minor point, I hope to raise my rating to '4: Borderline Accept', to indicate that the paper meets the acceptance level of NeurIPS.
> >
> > Again, thanks a lot for the kind and detailed response.
> >
> > Sincerely,

---

### Decision · Program_Chairs · 2025-09-17

**Decision:**

Accept (poster)

**Comment:**

The paper addresses the Generalized Category Discovery (GCD) task in federated settings. It is a new extension of federated learning. The proposed solution is reasonable with limited scientific contribution.

**strengths**

This work advances federated learning research by exploring a new and practical scenario.

The proposed solution is technically sound and well-supported by empirical analysis.

**weaknesses**

The design choices should be further discussed and better justified.

Given the composite nature of the proposed solution, a convergence analysis would strengthen the work by providing stronger theoretical guarantees.

**recommendation**

I recommend accepting this paper as an innovative contribution to federated GCD. It is expected that the authors will continue to refine this direction in future work, potentially providing a method with stronger theoretical convergence guarantees.

**reviewer consensus**

All reviewers gave positive scores for this work, with some reviewers increasing their scores after the rebuttal.